# Effects of Drip Irrigation and Fertilization Frequency on Yield, Water and Nitrogen Use Efficiency of Medium and Strong Gluten Wheat in the Huang-Huai-Hai Plain of China

Tianjia Hao, Zixin Zhu, Yulu Zhang, Shuai Liu, Yufan Xu, Xuexin Xu and Changxing Zhao *

Shandong Provincial Key Laboratory of Dryland Farming Technology, College of Agronomy, Qingdao Agricultural University, Qingdao 266109, China; haotianjia20230313@163.com (T.H.)
* Correspondence: zhaochangxing@126.com

**Abstract:** Drip irrigation can reduce water and fertilizer use; however, the frequency of topdressing required for drip irrigation for wheat in the Huang-Huai-Hai region is still unclear. Through two continuous wheat season field experiments, yield related traits under traditional surface irrigation (border irrigation) and three drip fertilization frequencies (DF2, DF3, DF4, that was, topdressing water and fertilizer twice, three or four times in the same way during the growth period) of three wheat cultivars (Jimai 22, Jimai 20, Shiluan 02-1) were studied. Increasing the frequency of drip irrigation fertilization could prolong the time of high-level photosynthesis, increase the dry matter distribution amount (DMDA) of stems and leaves, and add the weight of 1000 grains; it could increase the DMDA and nitrogen distribution amount (NDA) of the stems, leaves, and grains of Jimai 22, forming higher harvest index (HI) and nitrogen harvest index (NHI), but could reduce the DMDA of the grains of Jimai 20 and Shiluan 02-1, increasing NDA, reducing the harvest index but forming a higher nitrogen harvest index. The increase in drip irrigation fertilization frequency can improve protein content, increase grain number per spike, decrease spike number, improve the yield of medium gluten wheat, and improve nitrogen partial productivity and water use efficiency, while strong gluten wheat has a decrease in yield, nitrogen partial productivity, and water use efficiency. In summary, medium gluten wheat is more suitable for higher fertilization frequency in the Huang-Huai-Hai wheat region, while strong gluten wheat is the opposite.

**Keywords:** grain yield; dry matter; nitrogen uptake; drip irrigation; wheat



## 1. Introduction

Wheat production in China increased from 557 kg·ha$^{-1}$ in 1961 to 5810 kg·ha$^{-1}$ in 2021; the main reason for the increase in grain yield in recent years has been the increase in nitrogen application, but this has also caused environmental damage (National Bureau of Statistics of China, 1961–2021) [1,2]. The supply of water and fertilizer determines a further increase in production [1], but their excessive use may lead to high carbon emissions [3]. Thus, rational nitrogen application provides a way to regulate the physiological development of wheat [4]. Previous studies have shown that the nitrogen use efficiency (NUE) of wheat can be assessed using nitrogen fertilizer consumed per unit grain yield (PFPN) [1]. Determining appropriate nitrogen application rates are required to further tap the potential of increasing yield and nitrogen use efficiency.

The average annual rainfall in the Huang-Huai-Hai region is 500–600 mm, and the average annual temperature is 14.5 °C [5]. The Huang-Huai-Hai Plain has less annual rainfall in China [6], limited water resources [7], and at least 80% of the surface and underground water are used for agricultural irrigation [8]. Irrigation practice and infrastructure investment are relatively low [1], and there is an urgent need to take cultivation measures to improve agricultural water efficiency (WUE) and produce more food with less water [1,9].

Crop grain yield is closely related to dry matter accumulation and distribution [10,11], photosynthetic characteristics [10] as well as water and nitrogen utilization efficiency [11,12]. The dry matter in grains mainly originates from the vegetative organs [13,14]. Reasonable use of water and nitrogen can promote the growth of vegetative organs such as stems and leaves, maintain plant photosynthesis and nutrient transport, promote nitrogen uptake in grains, and increase protein content [13–15]. Balancing nitrogen supply and yield demands is the key to improving NUE [16]. Photosynthetic characteristics are significantly affected by soil water content and nitrogen fertilizer levels [17,18]. Water shortage will shorten the photosynthetic time and accelerate flag leaf senescence [19,20], but it may facilitate the degradation of stored substances in the plant and enhance the retransfer of the pre-flowering accumulation of dry matter [19]. Maintaining proper soil moisture content and higher nitrogen fertilizer levels can increase net photosynthetic rates (Pn), enhance grain filling capacity, and increase grain weight [18,20,21].

There have been numerous reports on optimal nitrogen application rates [4,12], drip irrigation [14,22], and irrigation frequency [13,23], but there are few reports on the frequency of drip irrigation fertilization. In this study, different medium and strong gluten wheat varieties were selected, and different fertilization frequencies were set through drip irrigation equipment under appropriate nitrogen application rates. The purpose was to: (1) determine the effects of drip irrigation topdressing frequency on yield, protein content, the distribution and accumulation of dry matter, and the distribution and uptake of nitrogen; (2) study the effects of fractional topdressing on photosynthetic characteristics and water and nitrogen use efficiency of medium gluten and strong gluten wheat; (3) optimize the topdressing frequency for medium gluten and strong gluten wheat in the Huang-Huai-Hai wheat region.

## 2. Materials and Methods

### 2.1. Experimental Site and Design

The field experiment of two wheat seasons was conducted in the Modern Agricultural Science and Technology Demonstration Park (Shandong province, China; 35.53°/N, 119.58°/E) of Qingdao Agricultural University from 2020 to 2022. The area belongs to a temperature monsoon climate (Table 1), with the main planting system being wheat corn rotation and the soil being mortar black soil (Figure 1). Medium gluten (Jimai 22) and strong gluten (Jimai 20, Shiluan 02-1) were selected and sown together on 11 October 2020 and 28 October 2021. The basic seedlings were about 2.2 million·ha$^{-1}$, and 90 kg·ha$^{-1}$ of $P_2O_5$ and $K_2O$ were applied; the area of each trial plot was 200 m$^2$, and the row spacing of wheat was 20 cm. This experiment used a drip irrigation system with emitters.All three rows of wheat were equipped with a capillary tube with a spacing of 60 cm. The distance between the emitter of each capillary tube was 300 mm, with a flow rate of 2 L/h and a diameter of 16 mm.

**Table 1.** Foundation soil capacity of 0–20 cm soil in the test area.

| Growing Season | Organic Matter (g·kg$^{-1}$) | Soil pH | Hydrolysable N (mg·kg$^{-1}$) | Available P (mg·kg$^{-1}$) | Available K (mg·kg$^{-1}$) |
|---|---|---|---|---|---|
| 2020–2021 | 16.21 | 7.55 | 127.9 | 15.14 | 136.5 |
| 2021–2022 | 17.24 | 7.62 | 129.3 | 15.98 | 134.2 |

The three varieties were medium gluten and strong gluten wheat varieties with relatively high yield and good quality selected previously. Under the premise of 210 kg·ha$^{-1}$ of total nitrogen application and 90 kg·ha$^{-1}$ of seed manure, the irrigation volume was measured by a water meter.

Traditional surface irrigation (border irrigation) (120 kg·ha$^{-1}$ of N fertilizer) was sprayed at the jointing stage. 75 mm irrigation was applied at the jointing stage and flowering stage respectively. Nitrogen application by drip irrigation was accomplished

through melting urea into water and injecting it into the field with drip irrigation facilities. Three kinds of frequency of fertilization under drip irrigation were set: Drip irrigation and topdressing twice during the entire growth period (DF2: 60 kg·ha$^{-1}$ of N fertilizer and 60 mm irrigation were applied at the jointing stage, and the same was true at the flowering stage); Drip irrigation and topdressing three times during the entire growth period (DF3: 40 kg·ha$^{-1}$ of N fertilizer and 40 mm irrigation were applied in drip irrigation at the jointing stage, and the same was true at the flowering and filling stages); Drip irrigation and topdressing four times during the entire growth period (DF4: 30 kg·ha$^{-1}$ and irrigation 30 mm of water were applied in drip irrigation at the jointing stage, and the same was true at the booting stage, flowering stage, and filling stage) (Tables 2 and 3).

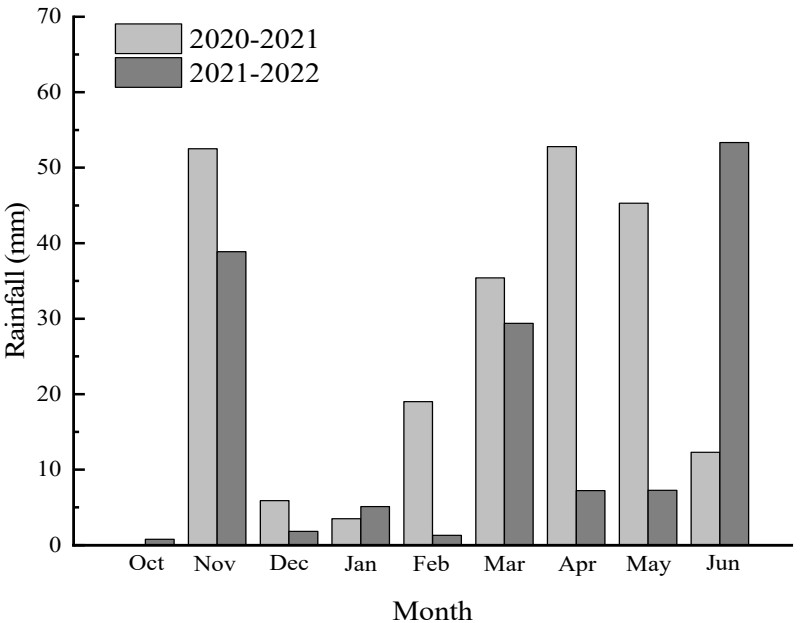

**Figure 1.** Rainfall in different months of 2020 to 2022. Note: The abscissa is the month of wheat growth period, and the ordinate is the monthly rainfall. Winter wheat in the Huang-Huai-Hai Plain is sown in October and harvested in June of the next year.

**Table 2.** Dates of key agronomic measures during wheat growth period.

| Wheat Seasons | Dates of Sowing | Dates of Harvest | Dates of Drip Fertigation | | | |
|---|---|---|---|---|---|---|
| | | | Jointing | Booting | Flowering | Grouting |
| 2020–2021 | 11 October | 17 June | 6 April | 25 April | 7 May | 25 May |
| 2021–2022 | 28 October | 18 June | 18 April | 21 April | 13 May | 27 May |

**Table 3.** Irrigation volumes(mm) and N application(kg·ha$^{-1}$) during wheat growth period.

| Amount of Irrigation and N Fertilizer | Irrigation Method | Treatment | Wheat Growth Period | | | | | |
|---|---|---|---|---|---|---|---|---|
| | | | Seeding | Jointing | Booting | Flowering | Filling | Total |
| N application (kg·ha$^{-1}$) | Border irrigation | CK | 90 | 120 | | | | 210 |
| | Drip fertigation | DF4 | 90 | 30 | 30 | 30 | 30 | 210 |
| | | DF3 | 90 | 40 | | 40 | 40 | 210 |
| | | DF2 | 90 | 60 | | 60 | | 210 |
| Irrigation volumes (mm) | Border irrigation | CK | | 75 | | 75 | | 150 |
| | Drip fertigation | DF4 | | 30 | 30 | 30 | 30 | 120 |
| | | DF3 | | 40 | | 40 | 40 | 120 |
| | | DF2 | | 60 | | 60 | | 120 |

*2.2. Sampling and Measurements*

2.2.1. Net Photosynthetic Rate (Pn), Stomatal Conductance (Gs), Transpiration Rate

The net photosynthetic rate (Pn), stomatal conductance (Gs), and transpiration rate (Tr) of wheat flag leaves were measured with the CIRAS-3 Portable Photosynthesis System at 9:00–11:00 a.m. and 2:00–4:00 p.m. every 7 days after flowering. Three leaves were measured for each treatment three times (nine in total).

2.2.2. Dry Matter Accumulation and Distribution, Nitrogen Uptake and Nitrogen Utilization Efficiency

At maturity, 30 aboveground plants of wheat were taken and divided into upper leaves, other leaves (the leaves on the wheat plant except the top three leaves), stems, grains, glumes + rachis, and remaining tillers. After being green at 105 °C for 30 min, they were dried at 75 °C to constant weight. A Kjeldahl automatic nitrogen determination instrument (Gerhardf) was used to measure nitrogen content, and this was repeated three times for each plot. The nitrogen utilization efficiency was represented by partial productivity of wheat nitrogen fertilizer [1]. The calculation formulas were as follows:

$$HI = (GY_M/DM_a) \times 100\% \tag{1}$$

$$GPS = (GY_M/DM_S) \times 100\% \tag{2}$$

$$GPC = GN_C \times 5.7 \tag{3}$$

$$GPY = GPC \times GY \tag{4}$$

$$PFPN = GY/N \tag{5}$$

$$NHI = GN_a/N_a \tag{6}$$

where HI (%) is the harvest index, $GY_M$ (kg·ha$^{-1}$) is the dry weight of grains at maturity, $DM_a$ (kg·ha$^{-1}$) is the dry matter accumulation at maturity, GPS (%) is the ratio of grains to ears, $DM_s$ (kg·ha$^{-1}$) is the dry weight of ears, GPC (%) is the protein content, 5.7 is the conversion coefficient of grain nitrogen and protein content, GPY (kg·ha$^{-1}$) is the protein yield, GY (kg·ha$^{-1}$) is the wheat yield, PFPN (kg·kg$^{-1}$) is the partial productivity of wheat nitrogen fertilizer, N (kg·ha$^{-1}$) is the total nitrogen application amount, NHI (kg·kg$^{-1}$) is the nitrogen harvest index, $GN_a$ (kg·ha$^{-1}$) is the nitrogen accumulation amount in grains, and $N_a$ (kg·ha$^{-1}$) is the nitrogen accumulation amount above ground.

2.2.3. Water Use Efficiency

Evapotranspiration (ET) in the whole growth period of each treatment was calculated as follows:

$$ET = I + P + G - D - \Delta S \tag{7}$$

where ET (mm) is the evapotranspiration during the whole growth period, I (mm) is the irrigation water volume, P (mm) is the precipitation, and $\Delta S$ (mm) is the change in soil water storage. In this experiment, G (mm) groundwater recharge, R (mm) surface runoff, and D (mm) infiltration below the root zone of crops could be ignored [24].

The water use efficiency (WUE) of wheat from sowing to maturity was calculated as follows [9]:

$$WUE = GY/ET \tag{8}$$

where WUE (kg·ha$^{-1}$·mm$^{-1}$) is water use efficiency, GY (kg·ha$^{-1}$) is wheat yield, and ET (mm) is evapotranspiration during the whole growth period.

2.2.4. Grain Yield

Before harvest, the number of wheat ears was measured (by converting the number of wheat ears in 1 m three rows), as well as the number of grains per ear (the average number of grains in 30 wheat ears). When harvesting, a small harvester was used to harvest

10 m, with an area of 14 m², and this was repeated three times. The wheat yield with 13% water content (13% water content was converted by drying 100 g grains to calculate the water content) and the 1000 grain weight (the total dry weight of 1000 wheat grains) were measured.

*2.3. Statistical Analysis*

Excel was used to collate the data, SPSS was used to perform the ANOVA of two factor randomized blocks, and the difference was compared at the 0.05 level (Duncan). Origin2019b was used to plot.

**3. Results**
*3.1. Photosynthetic Characteristics*

The Pn (net photosynthetic rate), Gs (stomatal conductance), and Tr (transpiration rate) of DF3 were affected by the water and fertilizer topdressing before flowering. At the flowering stage, the Pn, Gs, and Tr of DF3 were higher, while those for DF4 were lower. However, after increasing the frequency of water and fertilizer replenishment at the flowering stage and the filling stage (14 days after flowering), they were improved. In the late stage of grain filling, compared with the border irrigation CK, the treatment with more topdressing times could still maintain higher photosynthetic characteristics (Figure 2).

*3.2. Distribution and Accumulation of Dry Matter and Nitrogen*

The year*variety*Treatment had a very significant impact on the dry matter distribution proportion (DMDP) and nitrogen distribution amount (NDA) of other leaves (Tables S1 and S2). In addition, it had no significant impact on the allocation amount and proportion of dry matter and nitrogen in each organ. The dry matter distribution amount (DMDA) of the upper three leaves of wheat was negatively correlated with that of the grains ($p > 0.05$), while other leaves, stems, glume+rachiswere positively correlated with that of the grains ($p > 0.05$) (Figure 3A). This indicated that during the mature stage, more nutrients from the upper three leaves of wheat were transported to the ears, making the grains fuller, while more nutrients were left in other leaves, stems, glume+rachis. There was a significant negative correlation ($p < 0.05$) between the DMDP of wheat grains and the stems and leaves (Figure 3B). In the same plant, they formed a competitive relationship with the photosynthetic products produced by the leaves. The NDA of leaves and stems was negatively correlated with grains ($p > 0.05$), while the NDA of glumes and rachis was significantly positively correlated with grains ($p < 0.05$), indicating that plants with more NDA in the ear had less NDA in other parts (Figure 3C). The dry nitrogen distribution proportion (NDP) of grains, stems, and leaves showed a significant negative correlation ($p < 0.01$), while in the same plant, more nitrogen was concentrated in grains rather than other organs (Figure 3D). During the two years, the variety had a very significant impact on the allocation amount and proportion of stems and leaves during the mature period ($p < 0.01$) and had a significant impact on the allocation amount of grains ($p < 0.01$) (Table S1). The variety and treatment had a significant impact on the nitrogen allocation amount and proportion of leaves ($p < 0.01$) (Table S2). As the frequency of water and fertilizer increased, the DMDA of stems and leaves increased, and the grain allocation amount increased. However, within the same single plant of wheat, the DMDP of stems, leaves, and grains of Jimai 22 increased, while the DMDP of grains of Jimai 20 and Shiluan 02-1 decreased; the DMDP of stems and leaves was the opposite (Table S1). As the frequency of fertilization increased, the NDA of leaves, stems, and grains in Jimai 22 increased, while the NDA of glume+ rachis decreased. The NDA of leaves, stems, and grains in Jimai 20 and Shiluan 02-1 decreased, while the NDA of grains increased. In the same single plant of wheat, under higher water and fertilizer frequencies, the NDP of leaves, stems, glume+ rachis decreased, while the NDP of grains was on the opposite (Table S2).

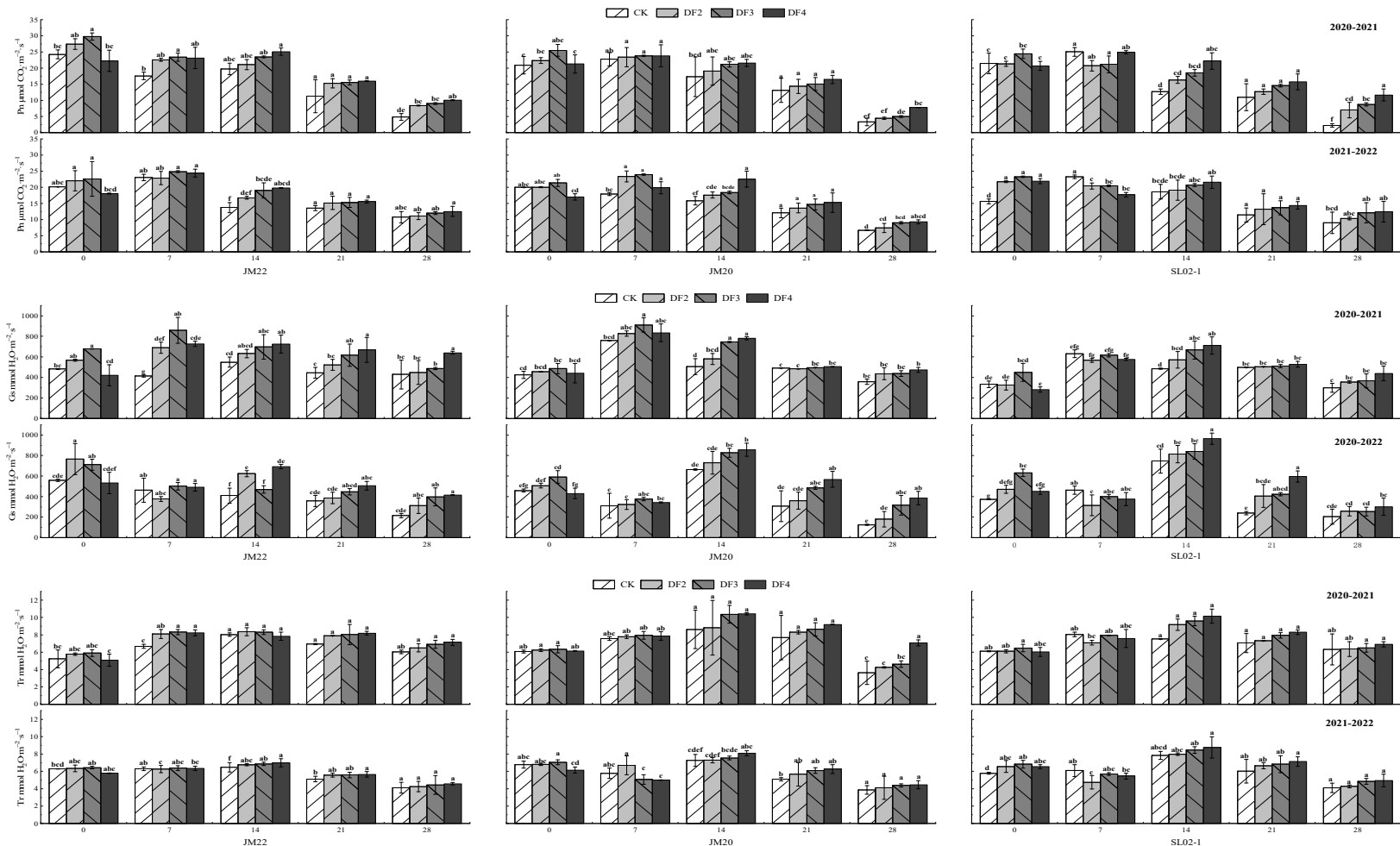

**Figure 2.** Net photosynthetic rate, transpiration rate and stomatal conductance after anthesis. Note: different lowercase letters represent significant differences at 0.05 level. The same below.

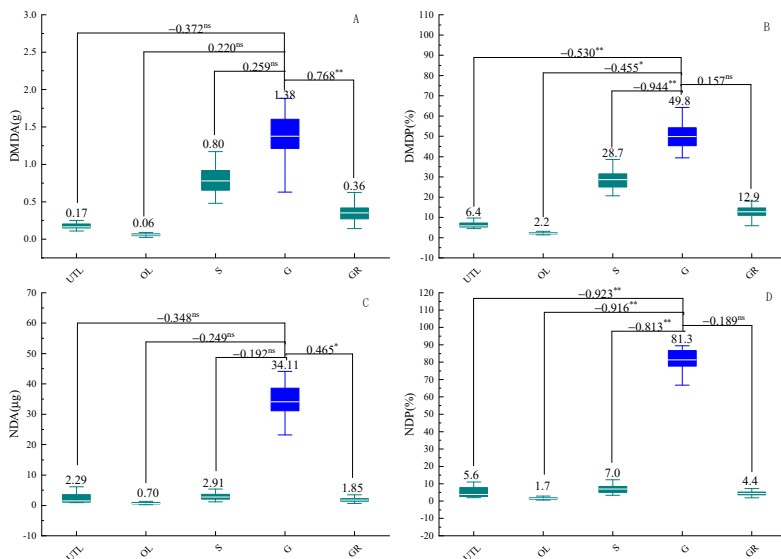

**Figure 3.** Distribution and proportion of dry matter in various organs of wheat. Note: DMDA is the distribution of dry matter in each organ, DMDP is the proportion of dry matter distribution in each organ, NDA is the amount of nitrogen uptake of each organ, and NDP is the proportion of nitrogen uptake of each organ; UTL is the upper three leaves, OL is the other leaves, S is the stem, G is the grain, and GR is the glume and rachis. (**A**–**D**) represent the DMDA, DMDP, NDA, and NDP of various organs in wheat, respectively. In (**A**), 0.17 g is the average of the upper three leaves, −0.372 is the Pearson correlation coefficient of UTL/G, * represents a significant correlation at the 0.05 level, ** represents a significant correlation at the 0.01 level, and ns represents no significant correlation, the same below.

Varieties and treatments had extremely significant effects on DMA (the dry matter accumulation; the weight of wheat organs per hectare) and NA (the dry matter uptake; the nitrogen weight of wheat organ per hectare) of upper leaves, grains, remaining stems and total amount, respectively ($p < 0.01$), and year*variety*treatment had no significant effect on DMA and NA of all organs. The treatment*variety had a significant effect on the NA of grains ($p < 0.01$) (Table **??**). The total DMA and NA of Jimai 22 were the highest in DF4, followed by DF3, which increased with the increase in fertilization frequency. On the contrary, Jimai 20 and Shiluan 02-1 had lower CK than DF2. The higher frequency of fertilization (DF3 and DF4) made the stem DMA and NA of Jimai 22 higher, while Jimai 20 and Shiluan 02-1 were the opposite; however, a similarity between the three varieties was that with the increase in fertilization frequency, the total DMA and NA of grains were higher (DF4 > DF3 > DF2 > CK).

### *3.3. Grain Yield*

Over two years, the variety*treatment (interaction between varieties and treatments) had a significant impact on the yield ($p < 0.01$), and both variety and treatment had a significant impact on the three elements of yield ($p < 0.01$) (Table 5). Under the appropriate amount of nitrogen and irrigation, the yield of medium gluten wheat Jimai 22 increased, while that of strong gluten wheat Jimai 20 and Shiluan 02-1 decreased with the increase in drip irrigation and topdressing times. The increase in water and fertilizer frequency resulted in fewer ears and increased grains per ear, and the 1000 grain weight of DF3 and DF4 was increased. The harvest index and grain/ear ratio of Jimai 22 increased with the increase in fertilization frequency, while Jimai 20 and Shiluan 02-1 were on the contrary. Compared with border irrigation, the yield of drip irrigation with twice topdressing increased slightly, the number of grains per ear and 1000 grain weight were higher, and the harvest index was lower.

**Table 4.** Dry matter accumulation and nitrogen absorption of various organs in wheat maturity.

| Year | Variety | Treatment | Upper Three Leaves DMA (kg·ha⁻¹) | NA (kg·ha⁻¹) | Other Leaves DMA (kg·ha⁻¹) | NA (kg·ha⁻¹) | Stem DMA (kg·ha⁻¹) | NA (kg·ha⁻¹) | Grain DMA (kg·ha⁻¹) | NA (kg·ha⁻¹) | Glume + Rachis DMA (kg·ha⁻¹) | NA (kg·ha⁻¹) | Remaining Tillers DMA (kg·ha⁻¹) | NA (kg·ha⁻¹) | Total DMA (kg·ha⁻¹) | NA (kg·ha⁻¹) |
|---|---|---|---|---|---|---|---|---|---|---|---|---|---|---|---|---|
| 2020–2021 | Jimai22 | CK | 1284 bcd | 26.7 bc | 452 b | 5.9 b | 5332 cd | 18.8 b | 7624 def | 176.9 b | 1783 ab | 10.7 bc | 68.8 ab | 8.9 ab | 17,164 c | 248 c |
| | | DF2 | 952 e | 19.3 c | 296 d | 4.9 bc | 5206 cd | 19.5 b | 8314 cd | 189.7 b | 2485 a | 17.3 a | 47.7 bc | 5.9 bc | 17,730 bc | 257 c |
| | | DF3 | 1111 de | 19.5 c | 350 cd | 5.8 b | 5940 ab | 22.0 ab | 9464 ab | 227.6 a | 2446 a | 15.2 ab | 54.0 bc | 6.6 abc | 19,851 a | 297 b |
| | | DF4 | 1177 cde | 19.9 c | 374 bcd | 6.1 b | 6342 a | 22.7 ab | 9940 a | 246.0 a | 2283 a | 14.1 ab | 36.4 bcd | 4.4 bcd | 20,481 a | 313 ab |
| | Jimai20 | CK | 1471 b | 23.1 bc | 565 a | 8.1 a | 6109 a | 28.4 a | 6744 fg | 169.8 b | 1429 b | 7.4 c | 67.4 ab | 8.7 ab | 16,993 c | 246 c |
| | | DF2 | 962 e | 8.2 d | 415 bc | 4.0 cd | 6392 a | 25.3 ab | 7610 def | 191.8 b | 2388 a | 13.0 ab | 90.3 a | 10.8 a | 18,670 b | 253 c |
| | | DF3 | 974 e | 8.1 d | 418 bc | 3.6 cd | 6157 a | 22.6 ab | 7295 efg | 189.6 b | 2585 a | 13.9 ab | 56.7 abc | 5.5 bcd | 17,997 bc | 243 c |
| | | DF4 | 1026 de | 7.6 d | 401 bc | 3.4 d | 6009 a | 20.2 ab | 6790 fg | 181.9 b | 2460 a | 13.1 ab | 39.2 bcd | 3.8 cd | 17,078 c | 230 c |
| | Shiluan02-1 | CK | 1795 a | 48.4 a | 358 cd | 8.4 a | 4609 e | 26.7 ab | 6552 g | 186.5 b | 1948 ab | 12.3 ab | 23.5 cd | 4.0 cd | 15,497 d | 286 b |
| | | DF2 | 1439 bc | 30.7 b | 346 cd | 6.4 b | 5515 bc | 26.6 ab | 8923 bc | 256.2 a | 1891 ab | 12.2 ab | 43.9 bcd | 6.9 abc | 18,554 b | 339 a |
| | | DF3 | 1404 bc | 28.9 b | 346 cd | 6.3 b | 5270 cd | 22.5 ab | 8333 cd | 240.5 a | 1962 ab | 12.4 ab | 27.2 cd | 3.8 cd | 17,587 bc | 315 ab |
| | | DF4 | 1453 b | 28.6 b | 354 cd | 6.4 b | 5010 de | 20.7 ab | 7806 de | 235.3 a | 2107 ab | 12.9 ab | 9.2 d | 1.3 d | 16,822 c | 305 b |
| 2021–2022 | Jimai22 | CK | 1093 bc | 9.8 b | 311 cd | 2.6 d | 4536 ab | 13.7 cd | 8395 e | 188 g | 2034 a | 7.6 b | 198 bc | 2.2 c | 16,567 f | 223 g |
| | | DF2 | 795 g | 5.9 d | 269 d | 2.5 d | 4358 ab | 12.1 cd | 9149 e | 205 fg | 2625 a | 10.1 b | 404 ab | 4.8 b | 17,601 ef | 240 fg |
| | | DF3 | 835 fg | 5.9 d | 315 cd | 2.7 d | 4526 ab | 11.0 d | 9285 de | 209 fg | 2593 a | 9.9 b | 224 bc | 2.5 c | 17,778 ef | 241 fg |
| | | DF4 | 868 fg | 5.8 d | 346 bcd | 2.5 d | 4686 ab | 11.3 d | 9487 de | 217 ef | 2487 a | 9.4 b | 125 c | 1.1 c | 17,998 de | 247 fg |
| | Jimai20 | CK | 1301 a | 10.1 b | 568 a | 5.1 ab | 4711 ab | 15.0 bc | 11163 abc | 247 cd | 2787 a | 12.2 ab | 307 bc | 1.4 c | 20,836 ab | 290 cd |
| | | DF2 | 1024 cd | 7.8 c | 445 b | 4.1 bc | 4812 ab | 15.1 bc | 11504 ab | 259 bc | 2633 a | 16.3 a | 580 a | 4.6 b | 20,997 ab | 307 bc |
| | | DF3 | 980 de | 6.2 d | 449 b | 3.5 cd | 4600 ab | 13.0 cd | 10375 cd | 239 cde | 2536 a | 11.6 ab | 242 bc | 1.7 c | 19,182 cd | 275 de |
| | | DF4 | 911 ef | 5.3 d | 425 b | 3.2 cd | 4209 b | 11.7 cd | 9252 de | 222 def | 2758 a | 11.5 ab | 209 bc | 1.5 c | 17,764 ef | 255 ef |
| | Shiluan02-1 | CK | 1312 a | 14.7 a | 385 bc | 5.1 ab | 4513 ab | 21.9 a | 8899 e | 224 def | 2144 a | 9.2 b | 406 ab | 4.4 b | 17,660 ef | 279 de |
| | | DF2 | 1100 bc | 14.7 a | 368 bcd | 5.2 a | 4971 a | 22.4 a | 12144 a | 305 a | 2424 a | 12.9 ab | 584 a | 6.6 ab | 21,590 a | 367 a |
| | | DF3 | 1137 b | 8.1 c | 399 bc | 3.5 cd | 4747 ab | 18.2 b | 11012 bc | 285 ab | 2564 a | 12.0 ab | 366 b | 2.6 c | 20,226 bc | 330 b |
| | | DF4 | 1086 bc | 6.6 d | 396 bc | 2.8 d | 4738 ab | 17.9 b | 10377 cd | 275 b | 2580 a | 10.8 ab | 304 bc | 2.0 c | 19,482 c | 315 bc |
| 2020–2022 | Year | | ** | ** | ns | ** | ** | ** | ** | ** | ** | * | ** | ** | ** | ns |
| | Variety | | ** | ** | ** | ** | ** | ** | * | ** | ns | ns | ** | ns | * | ** |
| | Treatment | | ** | ** | ** | ** | ns | * | ** | ** | ns | ** | ** | ** | ** | ** |
| | Year*Variety | | ** | ** | ** | ** | ** | ** | ** | ** | ns | ** | ** | ** | ** | ** |
| | Year*Treatment | | ns | ** | ns | ** | ns | ns | * | * | ns | ns | Ns | ns | ** | * |
| | Variety*Treatment | | ns | * | ** | ** | ** | ns | ** | ** | ns | ns | Ns | ns | ** | ** |
| | Year*Variety*Treatment | | ns | ns | ns | ns | ns | ns | ns | ns | ns | ns | Ns | ns | * | ns |

Note: DMA means the dry matter accumulation (the weight of wheat organs per hectare); NA means the dry matter absorption (the nitrogen weight of wheat organ per hectare). For the same column in the same year, different lowercase letters represent significant differences at 0.05 level, * represents significant differences at 0.05 level, ** represents extremely significant differences at 0.01 level, and ns represents insignificant differences at 0.05 level.

**Table 5.** Wheat yield and its three elements, harvest index and grain/ear ratio in two years.

| Year | Variety | Treatment | Spike Number ($10^4 \cdot ha^{-1}$) | Grain Number | 1000-Grain Weight | Yield ($kg \cdot ha^{-1}$) | HI (%) | GPS (%) |
|---|---|---|---|---|---|---|---|---|
| 2020–2021 | Jimai22 | CK | 614.4 ef | 31.00 bcd | 42.67 d | 7267 bcde | 44.4 ab | 81.0 abc |
| | | DF2 | 587.2 fg | 31.57 bc | 44.14 c | 7555 bc | 46.9 a | 77.1 abc |
| | | DF3 | 571.1 gh | 33.37 ab | 44.72 c | 7947 ab | 47.7 a | 79.5 abc |
| | | DF4 | 569.4 gh | 35.16 a | 44.20 c | 8545 a | 48.5 a | 81.3 ab |
| | Jimai20 | CK | 661.1 cd | 26.53 fg | 43.94 c | 7333 bcd | 39.7 c | 82.5 a |
| | | DF2 | 614.4 ef | 27.59 efg | 45.69 b | 7615 abc | 40.8 bc | 76.2 abc |
| | | DF3 | 586.7 fg | 27.72 efg | 46.32 b | 7234 bcde | 40.6 bc | 73.9 bc |
| | | DF4 | 545.0 h | 28.05 efg | 47.42 a | 6679 cdef | 39.8 c | 73.4 c |
| | Shiluan02-1 | CK | 855.6 a | 25.55 g | 31.10 g | 6082 f | 42.1 bc | 76.9 abc |
| | | DF2 | 752.2 b | 27.47 efg | 33.13 f | 6535 def | 48.1 a | 82.5 a |
| | | DF3 | 688.9 c | 28.59 def | 34.37 e | 6293 ef | 47.4 a | 81.0 abc |
| | | DF4 | 640.0 de | 30.09 cde | 34.27 e | 5109 g | 46.4 a | 79.0 abc |
| 2021–2022 | Jimai22 | CK | 596.1 de | 40.75 abc | 38.25 c | 9973 bcd | 51.0 a | 81.1 a |
| | | DF2 | 572.2 de | 40.80 abc | 41.03 abc | 10,045 bcd | 52.0 a | 77.8 a |
| | | DF3 | 561.1 e | 42.30 ab | 41.21 ab | 10,334 abcd | 52.2 a | 78.2 a |
| | | DF4 | 526.1 e | 44.22 a | 42.75 a | 11,176 a | 52.7 a | 79.3 a |
| | Jimai20 | CK | 786.7 b | 32.83 e | 39.55 bc | 10,469 abc | 53.6 a | 80.0 a |
| | | DF2 | 747.2 bc | 37.17 cd | 38.78 bc | 10,657 ab | 54.8 a | 81.4 a |
| | | DF3 | 633.9 d | 37.75 cd | 40.81 abc | 10,078 bcd | 54.1 a | 80.5 a |
| | | DF4 | 550.9 e | 38.10 cd | 41.21 ab | 8917 e | 52.1 a | 78.2 a |
| | Shiluan02-1 | CK | 898.8 a | 34.50 de | 30.09 e | 9854 bcd | 50.4 a | 80.5 a |
| | | DF2 | 896.7 a | 35.82 de | 30.03 e | 10,105 bcd | 56.3 a | 83.4 a |
| | | DF3 | 784.4 b | 35.67 de | 33.17 d | 9572 cde | 54.5 a | 81.1 a |
| | | DF4 | 707.2 c | 38.33 bcd | 33.23 d | 9485 de | 53.3 a | 80.1 a |
| 2020–2022 | Year | | ** | ** | ** | ** | ** | ns |
| | Variety | | ** | ** | ** | ** | ** | ns |
| | Treatment | | ** | ** | ** | ns | * | ns |
| | Year*Variety | | ** | ns | ** | ** | ** | ns |
| | Year*Treatment | | ** | ns | Ns | ns | ns | ns |
| | Variety*Treatment | | ** | ns | Ns | ** | ns | ns |
| | Year*Variety*Treatment | | * | ns | Ns | ns | ns | ns |

Note: CK refers to border irrigation, and DF2, DF3, and DF4 represent the topdressing of nitrogen fertilizer by drip irrigation for 2, 3, and 4 times respectively. JM22, JM20, and SL02-1, respectively, represent the abbreviations of wheat varieties Jimai22, Jimai20, and Shiluan02-1. For the same column in the same year, different letters represent significant differences at 0.05 level, * represents significant differences at 0.05 level, ** represents extremely significant differences at 0.01 level, and ns represents insignificant differences at 0.05 level.

### 3.4. Protein Content and Protein Yield

The protein content of border irrigation was relatively low, and as the frequency of fertilization increased, the protein content of the three varieties increased accordingly (Figure 4). In the first year, the DF4 of Jimai 22, Jimai 20, and Shiluan 02-1 significantly increased by 8.6%, 6.4%, and 4.9% compared to DF2, respectively. In the second year, there was no significant difference in protein content between different fertilization frequencies. The DF4 protein yield of Jimai 22 increased significantly by 22.8% and 13.5% compared to DF2 in two years, while Jimai 20 decreased significantly by 6.7% and 11%, respectively. Shiluan02-1 in the first year decreased significantly by 18%, and there was no significant difference in the second year.

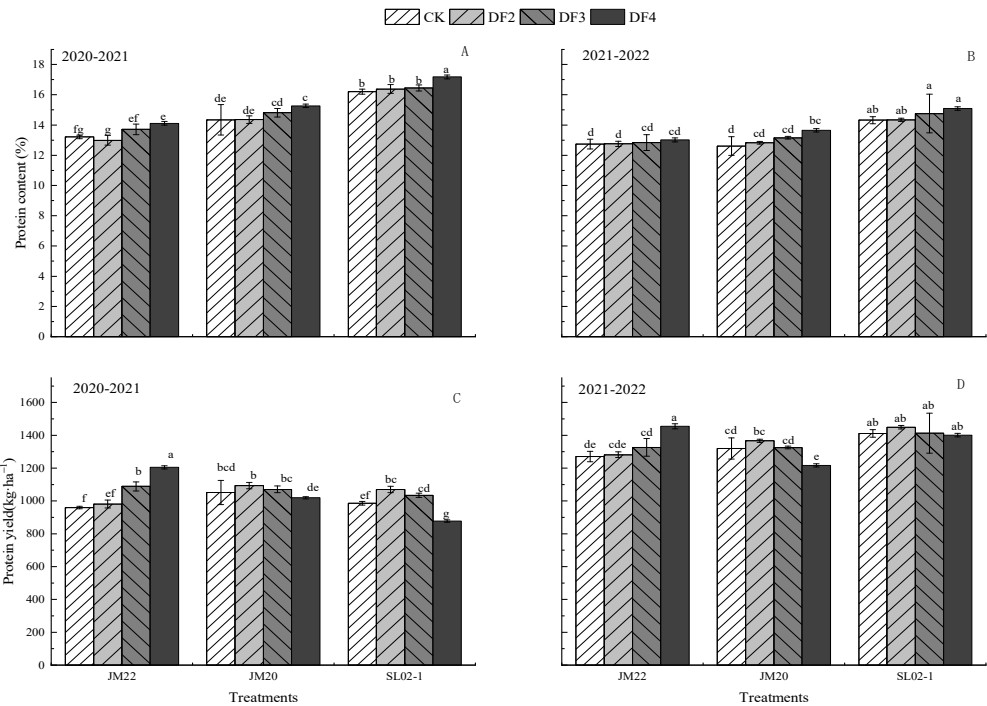

**Figure 4.** Wheat protein content and yield. Note: CK refers to border irrigation; DF2, DF3, and DF4 represent the topdressing of nitrogen fertilizer by drip irrigation for 2, 3, and 4 times respectively. JM22, JM20, and SL02-1, respectively, represent the abbreviations of wheat varieties Jimai22, Jimai20, and Shiluan02-1. (**A**) shows the protein content of different drip irrigation fertilization frequencies in the first year, while (**B**) shows the protein content in the second year. (**C**) shows the protein yield of different drip irrigation fertilization frequencies in the first year, while (**D**) shows the protein yield in the second year. Different lowercase letters represent significant differences at 0.05 level (Duncan).

### 3.5. Water and Nitrogen Utilization Efficiency

With the increase in fertilization frequency, the NHI (nitrogen harvest index) of the three varieties increased (DF4 > DF3 > DF2 > CK) (Figure 5). Over the past two years, with the increase in nitrogen application frequency, the nitrogen fertilizer partial productivity (PFPN) of Jimai 22 increased. There was no significant difference in the first year, and DF4 significantly increased by 11.3% compared to DF2 in the second year. Water use efficiency (WUE) showed an increasing trend, but there was no significant difference between different topdressing frequencies. The PFPN and WUE of Jimai 20 and Shiluan 02-1 increased with the increase in fertilization frequency. In the second year, the PFPN and WUE of DF4 in Jimai 20 were significantly reduced by 19.5% and 22.2%, respectively, compared to DF2. In the first year, the PFPN and WUE of DF4 from Shi Luan 02-1 were significantly reduced by 27.9% and 34.5%, respectively, compared to DF2.

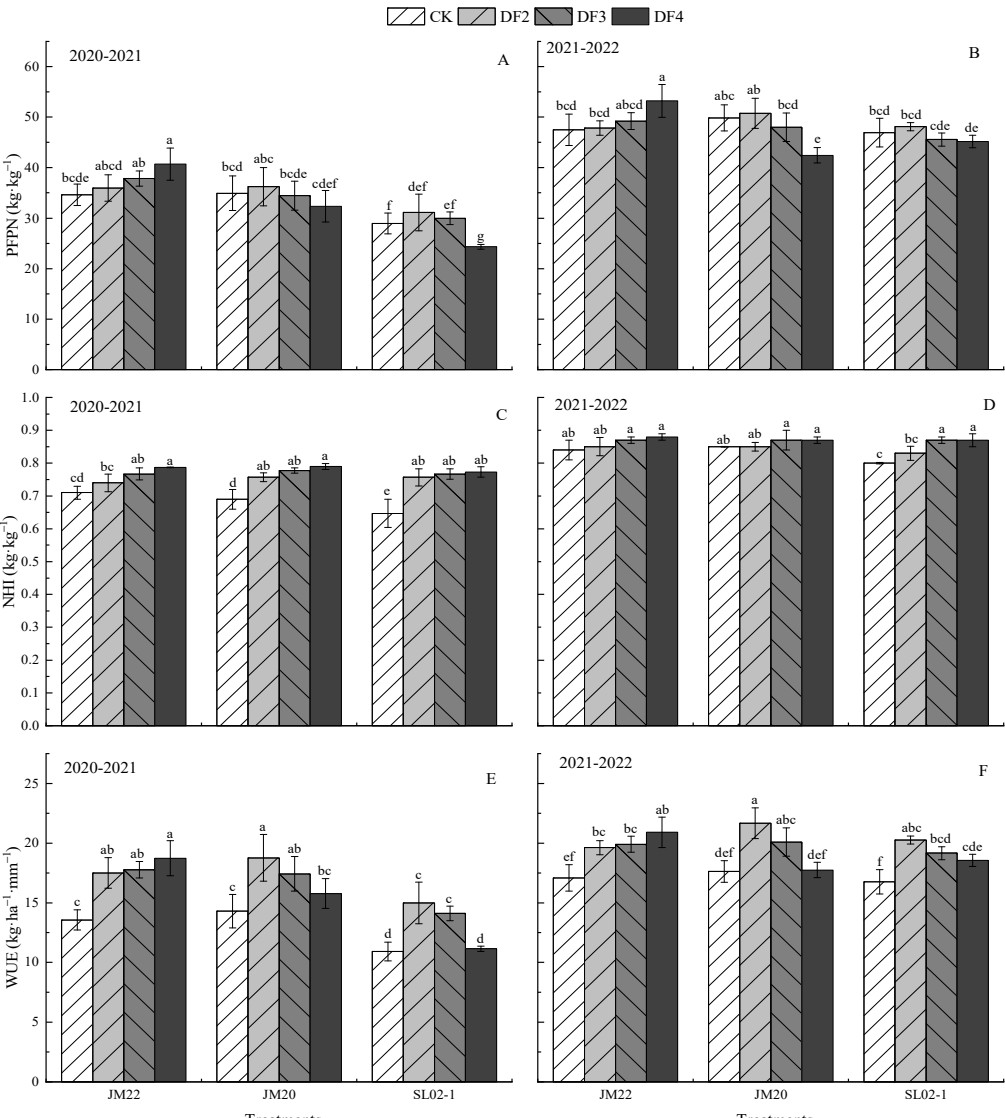

**Figure 5.** Water and nitrogen use efficiency of wheat under different drip irrigation and topdressing frequencies. Note: (**A**) shows the PFPN of different drip irrigation fertilization frequencies in the first year, while (**B**) shows the PFPN in the second year. (**C**) shows the NHI of different drip irrigation fertilization frequencies in the first year, while (**D**) shows the NHI in the second year. (**E**) shows the WUE of different drip irrigation fertilization frequencies in the first year, while (**F**) shows the WUE in the second year. Different lowercase letters represent significant differences at 0.05 level (Duncan).

## 4. Discussion

The Gs value is the surface conductivity of $CO_2$ and $H_2O$ [25]. The increase in the Gs value indicates that the assimilation of $CO_2$ increases slightly and the Tr increases slightly, leading to a higher Pn [26]. Previous studies show that under the influence of nitrogen, flag leaf photosynthesis contributes less to yield, has no positive correlation, and is greatly affected by the weather [18]. Some studies also show that nitrogen supplementation in the late growth period can prolong the high level of photosynthetic time [27], delay leaf senescence [28], and increase the Pn, Gs, and Tr, which are beneficial to grain filling and yield [27]. There is a very significant positive correlation between grain filling and 1000 grain weight [29]. In this study, the increase in fertilization frequency delayed flag leaf senescence, prolonged high-level photosynthesis, and increased the 1000-grain weight and grain nitrogen uptake, thereby increasing the NHI and protein content. The increase in

DMDP in the grains of medium gluten wheat Jimai 22 resulted in a higher HI for DF4, but the DMDP in the grains of strong gluten wheat Jimai 20 and Shiluan 02-1 decreased, resulting in a lower HI. Ruiqi Ma et al. pointed out that the changes in the distribution proportion of dry matter in wheat with different gluten types are not entirely the same. When the DMDP of strong gluten wheat grains decreases, the DMDP of leaves increases [30]. Previous studies have shown that the distribution and proportion of dry matter and nitrogen affect the total DMA and NA, the dry matter quality of each organ of wheat is shown as grain > stem > glume+ rachis > leaves at maturity [30]. In our study, the distribution, DMDA and DMDP of each organ was shown as grain > stem > glume + rachis > upper three leaves > other leaves.

Previous studies have shown that the DMA has a significant positive correlation with the yield and HI [11]. The yield is significantly affected by the transport of dry matter before anthesis and the DMA after anthesis [14] and by water, fertilizer, and weather conditions [13,31,32]. The fertilization period is particularly important. Different fertilization periods affect the yield and DMA [15]. Different varieties are affected differently by the fertilization period, leading to the differences in DMA and NA between yield at maturity [15,23,33]. The study by Layeth Moradi et al. showed that increasing the frequency of drip irrigation increased the DMDA of mature stems, leaves, and grains [13]. In this study, an increase in the DMDA of stems, leaves, and grains was also observed by increasing the frequency of drip irrigation fertilization. The total DMA and NA of Jimai 22 in medium gluten wheat increased with the increase in fertilization frequency, while strong gluten wheat Jimai 20 and Shi Luan 02-1 showed the opposite effect. This may be affected by the number of ears. Although the DMDA of DF2 grains in strong gluten wheat was lower, the number of ears was higher, affecting the total DMA. Moreover, Huifang Han et al. pointed out that increasing the frequency of irrigation and delaying irrigation will reduce dry matter accumulation [23]. Layeth Moradi et al. believe that increasing the frequency of irrigation can increase yield and dry matter accumulation [13], which may be influenced by the variety and external environment.

Drip irrigation helps to save water and fertilizer when growing wheat. Our previous research showed that optimal nitrogen application was 210 kg·ha$^{-1}$ [22]. Much research has been conducted on phased irrigation or fertilization [15,23,34]. The yield is determined by three factors (grain number, 1000 grain weight, spiker number). Irrigation [23] and fertilization [15,34] at the jointing stage are beneficial to increase the number of ears. The 1000 grain weight increases with the extension of the nitrogen topdressing period [33,35]. There is an antagonistic relationship between the 1000 grain weight and the number of grains per unit area; medium gluten wheat is more dependent than strong gluten wheat on late fertilization [34]. The results showed that with the increase in drip fertilization frequency, the 1000 grain weight increased, the number of grains per spike increased, and the number of spikes decreased; the yield and harvest index of strong gluten wheat with two times of drip irrigation and topdressing (jointing stage and flowering stage) were the highest. With the increase in fertilization frequency, the grain yield, protein yield, and harvest index decreased. The yield and harvest index of medium gluten wheat under four drip irrigation treatments were the highest. With the increase of fertilization frequency, the yield, protein yield and harvest index decreased.

Previous studies have shown that PFPN (defined as the ratio of yield to nitrogen application) is an important indicator to measure the contribution of nitrogen to yields [1]. A higher yield is usually attributed to a higher nitrogen uptake and nitrogen use efficiency of shoots [36,37]. In this study, a higher yield also had a higher PFPN. Previous studies show that the WUE of strong gluten wheat irrigated twice is the highest [38]. This study showed that the water use efficiency of medium gluten wheat increased with the increase in fertilization frequency, while that of strong gluten wheat was the opposite.



## 5. Conclusions

In the middle and late filling stage, the PN, GS, and tr of DF4 were higher than those of df2, and the grain weight and DMDP were higher. After increasing the frequency of drip irrigation fertilization, the duration of high-level photosynthesis was prolonged, the DMDA of stems and leaves was increased, and the weight of 1000 grains was increased. This increased the DMDA and NDA of stems, leaves, and grains of Jimai 22, resulting in an increase in HI and NHI. The DMDA and NDA of grains of Jimai 20 and Shiluan 02-1 decreased, resulting in a smaller HI but a higher NHI; increasing the frequency of drip irrigation fertilization can increase protein content; as the frequency of drip irrigation fertilization increased, the number of grains per spike of medium gluten wheat increased and the number of spikes of medium gluten wheat decreased, resulting in an increase in yield, nitrogen partial productivity, and water use efficiency for medium gluten wheat, while for strong gluten wheat, the opposite was true.

**Supplementary Materials:** The following supporting information can be downloaded at: https://www.mdpi.com/article/10.3390/agronomy13061564/s1, Table S1: Dry matter distribution amount and proportion of each organ in wheat maturity; Table S2: Nitrogen distribution amount and proportion in wheat organs.

**Author Contributions:** Methodology, T.H.; Investigation, T.H., Z.Z., Y.Z., S.L., Y.X. and X.X.; Data analysis, T.H.; Writing—original draft preparation, T.H; Writing—review and editing, T.H. and C.Z.; Funding acquisition, C.Z. All authors have read and agreed to the published version of the manuscript.

**Funding:** This work was supported by the Major scientific and technological innovation Project of Shandong Province (2019JZZY010716), the National Key Research and Development Program of China (2018YFD0300604), the Key Research and Development Project of Shandong Province (2022CXPT009) and the Key Industrial Projects to Replace Old and New Driving Forces in Shandong Province (2021-54).

**Data Availability Statement:** Data are contained within the article.

**Acknowledgments:** Thanks to Yingzhen Kong of Agronomy College of Qingdao Agricultural University and Malcolm O'neill from University of Georgia for their helps in the language polishing and revision suggestions of this research paper.

**Conflicts of Interest:** The authors declare no conflict of interest.

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
