# Peer review of "Effects of Drip Irrigation and Fertilization Frequency on Yield, Water and Nitrogen Use Efficiency of Medium and Strong Gluten Wheat in the Huang-Huai-Hai Plain of China"

_agronomy, doi:10.3390/agronomy13061564_

Round 1

Reviewer 1 Report

This manuscript investigates the effects of drip fertilization frequencies on medium and strong gluten wheat in Huang-Huai-Hai Plain of China. The paper format and data analysis need to be improved. The paper falls within the general scope of the journal. Please see below for specific comments.

Abstract

- Line 16The first appearance of the abbreviation needs to be explained.

- Line 23change efficiency .In to efficiency. In

Introduction

The focus of this paper is on the fertilization frequency and cultivars, which should be reflected in the introduction.

- Line 28: delete Fan et al believe that

- Line 31: Add a space between the citation and the main text. Modifications throughout the paper.

- Line 32: delete first ., change the second . to ,

- Line 39: less fertilizer?

- Line 41: This paragraph was poorly written and the theme is somewhat confusing. Please rewrite.

- Line 44-45: These two lines are duplicated.

- Line 50: delete a space.

- Line 51: This sentence has nothing to do with the topic of this paragraph.

- Line 54: This sentence should be moved to paragraph 2.

- Line 65: The study should be elicited from the perspective of the research question.

Materials and methods

- Line 76: delete (Table 1 and Figure 1), add this elsewhere.

- Line 80: how much P and K fertilizer?

- Line 82: Harvest information is placed in 2.3.4

- Line 91: 240?

- Line 95: Experimental treatments needs to be rewritten, which is a bit unclear.

- Line 97: unit error.

- Line 106: Add units in the table.

- Line 107: The table name is placed on top of the table.

Results 

- Line 168: were higher, which treatment?

- Line 177: delete space.

- Line 178: other leaves (Table ?).

- Line 181: delete this sentence.

- Line 222: change the treatment of variety* to the variety

- Line 223: add a space.

- Line 227: the same? Unclear expression.

- Line 230-234: Dont use the data in the table over and over again, comparisons between treatments are required.

Table 4 and 5: It is not necessary to analyze the response variables every year.

-3.4 part: If there is no significant difference between the treatments, there is no statistical difference, indication that no data comparison can be made. Modifications throughout the paper.

- Line 262: by 2%? however, there was no significant difference between the two treatments.

- Line 267: on the contrary? however, there was no significant difference in protein yield among frequency treatments in 2021-2022 (Fig. 4).

-3.5 part: similar to the problem in the previous section.

Discussion

- Line 307-309: what is the reason for the difference between varieties?

- Line 334: Is there literature to support this?

- Line 353: The presentation is inconsistent with the data.

- Line 355:  It is enough for the abbreviation to be explained once in the text.

- Line 358: delete the second sentence.

- Line 362: after increasing irrigation frequency?

Add some results about the treatments to the conclusion section.

- Line 379: Wrong name.

References

 - Lines 395Abbreviation for the journal.

 - Lines 404Note the italics.

 - Lines 418Note the order of first and last names.

 - Lines 420: Wrong name.

 - Lines 436: Page numbers are missing.

no

Reviewer 2 Report

The paper entitled "Effects of drip irrigation and fertilization frequency on yield, water and nitrogen use efficiency of medium and strong gluten wheat in the Huang-Huai-Hai Plain of China" contains interesting research results. However, I have a few comments:

The abstract uses the abbreviation MDMA, NDA. You should first give its full name before using it, even if the abbreviation is obvious.

Please replace the word increase with a synonym in the abstract because it occurs too often.

Line 32: the full stop before and immediately after [1] is unnecessary.

Please always put a space before a quotation bracket e.g. [5].

Line 61, 63: do not use the phrase,,we" - write impersonally.

In the description of the place of the experiment, please state the country - this is currently missing. Please also elaborate on the site's description: what is the kilter there, what is the multi-year rainfall and temperatures, what crops dominate etc....

Fig.1: the word rainfall is written together as one word.

Fig2: Descriptions - letters denoting similarities and differences are too small. The graphic is currently illegible. Please correct it.

Line 231-232: are your previous studies published anywhere>? If so cite the reference.

The discussion is poorly written. The authors should relate their results to more previously published studies. I suggest the authors expand this chapter and add new aspects.

 The entire manuscript text should be decently checked and formatted for the journal's requirements. The text contains many unnecessary spaces, commas, full stops and typos. Please also do not use phrases such as we, our.

The level of English should be improved throughout the work. Currently, it is average. There is a lot of repetition, and sentences are poorly phrased or incomprehensible. I would ask the authors to improve the language of the work.
